# A Promising Predator-In-First Strategy to Control Western Corn Rootworm Population in Maize Fields

Antoine Pasquier [1,2,*], Lucie S. Monticelli [2], Adeline Moreau [1], Benjamin Kaltenbach [1], Candice Chabot [1], Thibault Andrieux [1], Maxime Ferrero [1] and Elodie Vercken [2]

1    Bioline Agrosciences, R&D, 1306 Route de Biot, 06560 Valbonne, France; adeline.moreau9@gmail.com (A.M.); kaltenbachbenjamin@gmail.com (B.K.); candice.cbt@gmail.com (C.C.); tandrieux@biolineagrosciences.fr (T.A.); maxime.ferrero@gmail.com (M.F.)

2    Université Côte d'Azur, INRAE, CNRS, UMR ISA, 06000 Nice, France; lucie.monticelli@gmail.com (L.S.M.); elodie.vercken@inrae.fr (E.V.)

\*    Correspondence: antoinepapass@gmail.com; Tel.: +33-648194148

**Abstract:** Western Corn Rootworm is a pest of maize that mostly damages roots. Many alternative strategies have been explored to control this species, with little or non-lasting success, and it remains a threat to maize production worldwide. *Gaeolaelaps aculeifer*, a soil-dwelling predatory mite that inhabits the first few centimeters of the soil, showed high predatory potential against WCR larvae in the laboratory. In this study, we explored the efficiency of *G. aculeifer* against WCR in more realistic contexts. First, we infested maize plants isolated in pots in a greenhouse with WCR, and tested the impact of different densities of mites on plant protection. Using standard indicators of WCR population presence and impact, we confirmed that *G. aculeifer* has the potential to control WCR at densities starting from 100 mites/plant. Then, considering that the release of a large amount of biocontrol agents at WCR emergence might be too costly and constraining for large-scale implementation, we tested the efficiency of a predator-in-first strategy in a maize field infested by WCR. The goal was to introduce fewer *G. aculeifer* combined with *Aleuroglyphus ovatus* eggs as an alternative food source in order to let the mite population grow in the field and reach sufficient density at the critical stage for protection. This strategy gave comparable results to pesticide on all indicators examined in our field trial, highlighting the potential to sustainably manage this pest.

**Keywords:** predatory mites; *Diabrotica virgifera virgifera*; biological control; *Gaeolaelaps aculeifer*; underground pests; predator-in-first

## 1. Introduction

With 1068 million tons produced in 2017 [1], Maize (*Zea Mays* L., 1753) is the first cereal produced in the world. The Western Corn Rootworm (WCR) (*Diabrotica virgifera virgifera*) LeConte, 1858, is a ground beetle originating from Central America [2,3] whose larvae inhabit the first centimeters of soil and consume Maize roots [4], which decreases water and nutrient uptake [5] and increases the risk of lodging and attack from plant pathogens [6]. Pupae complete their development in the soil while adults feed on the aerial part of the plant, such as silks. Although most economic losses are related to root damages by larvae, high densities of adult individuals can also affect plant reproductive organs, leading to a reduction in grain production and yield loss [7]. WCR is now established all over North America [8] and has been spreading rapidly in Europe since its first observation in Serbia in 1992. The lack of coevolution with native species prevents natural regulation of WCR by local predators, parasitoids or pathogens [9,10], thus resulting in huge economic losses. In 1986, the total cost resulting from yield decrease and pest management in North America was estimated over 1 billion USD [11].

Due to these important threats, major efforts were devoted to developing control strategies against WCR. These efforts were particularly challenged by the species' impor-

tant capacity to adapt to traditional pest management methods. Some populations became increasingly resistant to pesticides [12] and genetically-modified organisms (GMOs) [13]. Although the situation in Europe appears currently under control [14] due to culture rotation use promoted by European legislation [15], resistance to crop rotations has been observed in USA [16], and the demand for alternative control methods remains high. With this objective, many biological control solutions have been investigated, such as parasitoids [17–19], toxin from bacteria [20], entomopathogenic fungi [21,22] or nematodes [23–30] but none of these solutions proved suitable for large-scale use in maize fields.

Another promising, yet under-investigated strategy is the use of soil-dwelling predators of WCR larval stages. In particular, predatory mites are at the top of the soil trophic web [31]. Most species are mobile predators feeding on collembola, nematodes, insect larvae, insects eggs and other microarthropods [32,33]. Their effectiveness as biological control agents in regulating soil pest populations have contributed to their use both in the field and in the greenhouse [34,35]. A first test of their potential against WCR was done by [36], confirmed recently by Pasquier et al., [37], who tested the predation success of four mite species from four different genera on WCR in the laboratory. Among the different species tested, *Gaeolaelaps aculeifer* Canestrini (Arthropoda: Mesostigmata) showed the best results. *G. aculeifer* is a soil predatory mite used as a biological control agent in a greenhouse for controlling populations of nematodes, mealybugs, soil flies, and thrips pupae [32,38–40] in 17 European countries [41]. This cosmopolitan organism, naturally present in the soil [42] can move and burrow up to several tens of centimeters to find its prey. Ref. [43] highlight its capacity to overwinter and its resistance to low temperature. A laboratory experiment demonstrated the voracity of the species since in 1 out of 2 times, it is able to detect and consume prey of varying sizes (200 µm to 5 mm) in less than 10 min [37]. This work also showed that *G. aculeifer* does not feed on WCR eggs, meaning the biocontrol strategy needs to target the neonate stage.

In the present study, we first tested the potential of *G. aculeifer* to control WCR in a greenhouse. We released different densities of *G. aculeifer* around single maize plants infested with WCR at the estimated time of larvae emergence [44] to establish (a) whether predatory mites are able to find and eat WCR larvae directly in the soil (b) to estimate *G. aculeifer* density to control WCR populations at larvae emergence. However, to be applied efficiently in the field, this method of introduction needs to overcome several limitations: (i) introducing a high-density of predatory mites is particularly expensive; (ii) introducing a biocontrol agent at larvae emergence requires supplementary intervention for the farmer at 4–5 leaves stage; (iii) the vulnerable developmental stage, i.e., the WCR larvae emergence [37] is difficult to predict, may vary between environments, and may be missed when releasing predatory mites [45]. Based on all these constraints, we explored form the second time, another strategy in a field trial. The predators-in-first (PIF) strategy, first suggested by McMurtry et al. [46], aims to let the predators establish before the targetable pest or stage is present. This method relies on generalist predators' capacity to feed, survive and reproduce on an alternative food source provided by the environment [46–49]. This method showed its efficiency on predatory mites *Amblyseius swirskii* (Acari: Phytoseiidae) Athias-Henriot, 1962 previously introduced and fed with pollen to control thrips [50,51]. In this second experiment, we investigated the efficiency of a PIF in one maize field naturally infested by WCR at the egg stage, which is not targetable by predatory mites [37]. We introduced a smaller number of predatory mites (compared to the inundative strategy in the greenhouse experiment) early in the season (at sowing), combined with alternative food in order for the mite population to reach a sufficient density when WCR larvae, i.e., both the sensitive and the harmful stage, emerge. Indeed, WCR overwinters as eggs and emerge only when temperature increases at spring (Fisher, Hampton, NH, USA, 1987) and maize plants have grown enough to provide food for the larvae [8]. Because the delay between introduction (sowing) and larvae emergence is hard to predict [52], it was necessary to sustain the *G. aculeifer* population with an alternative food source during this period. We chose to use a known prey for *G. aculeifer*, *Aleuroglyphus ovatus* (Troupeau 1879) [53,54], which is also

used in our rearing unit at Bioline AgroSciences. To establish a proof of concept of this PIF strategy in realistic conditions before a wider experimental plan, we led the trial in one maize field and compared it with pesticide treatment and a negative control.

## 2. Material and Methods

### 2.1. Greenhouse Trial

2.1.1. Plant Material

The maize variety used in the greenhouse experiment is "Figaro", commercialized by the seed company "Semences de France". The seeds used were produced in 2019 and were not treated by the manufacturer. They were introduced at 5 cm depth into the soil and covered with soil.

2.1.2. Arthropods Production and Preparation

*Diabrotica virgifera virgifera*. *Diabrotica virgifera virgifera* eggs were provided by CABI (Centre for Agricultural Bioscience International) laboratory in Hungary, then stored for 22 days below their developmental threshold temperature, at $7 \pm 0.5$ °C. Eggs were separated from soil using metal sieves of 355 and 600 μm mesh. Eggs contained in the 355 μm mesh sieves were collected and inspected using an Olympus SZ 61 binocular microscope to separate viable eggs from damaged ones. Color and turgidity were used as criteria to distinguish viable, light-colured eggs from presumably damaged, dark-colured eggs. Viable eggs were collected with a polyamide-fibers brush, 1.3 mm (Ref: Pebeo Lotus Deco Round No.10-0) and used in the greenhouse trial.

*G. aculeifer.* In the greenhouse experiment, mites came from EWH Bioproduction, Tappernøje Denmark. The population was maintained during 8 months on a substrate made of 1/3 blond sphagnum peat and 2/3 of fine (0.5 to 2.8 mm) vermiculite and fed a mixture of all *A. ovatus* stages [37]. Predatory mites were stored in climatic chambers at $25 \pm 0.5$ °C and $70 \pm 10\%$ RH with a constant obscurity. A mixture of all stages of *A. ovatus* was used as food, and extra water was provided three times a week in a $100 \times 94 \times 60$ mm$^3$ bugdorm-5002 with 30 μm nylon screen port sold by Bugdorm© Megaview science corporation, Taiwan. Before application, *G. aculeifer* population sizes in the bugdorms were assessed. For doing so, a sample of the sphagnum/vermiculite was weighed and then sieved using a 500 μm metal sieve to remove substrate and a 250 μm metal sieve to retrieve mites. *G. aculeifer* individuals were then poured into 50 mL of distilled water then homogenized. Three samples of 2 mL of this mix were placed on a counting chamber under a SZ61 Olympus binocular microscope with a drop of surface-active agent to break the water surface tension and counted ($37 \pm 1.73$ mites/2 mL). *G. aculeifer* density in the rearing medium was then calculated, and Falcon tubes containing the required weight of substrate for each maize pot were prepared. At last, each Falcon tube containing the mix of growth substrate and *G. aculeifer* individuals was released close to the maize plant on the soil surface of each maize pot at the estimated time of WCR larvae emergence [44].

2.1.3. Experimental Setup

The experiment was performed in greenhouses at Bioline Agrosciences research site located in Valbonne, France, in 2019. Greenhouses ventilation and cooling parameters were set not to exceed 30 °C (Range 15 °C–30 °C). The cooling system is made of a ventilation spreading water inside the greenhouse reducing temperature as well as increasing humidity (~60%). The experiment to test the effect of *G. aculeifer* density to control the WCR population was performed in a compartment (48 m$^2$) with 6 rows composed of ten maize plants (60 plants in total). Each maize plant grew up isolated in a black plastic 50-L pot with a top-diameter of 46 cm and a bottom-diameter of 39 cm. Each plant was watered with 1 L of water each day except on Fridays where 2 L were poured in the pots and no extra water was added until Monday. Three 25 mm diameter holes were drilled at 4.5 cm of the bottom of each pot, to make sure no excess of water would result from plant watering. 50 L of topsoil were used as growth substrate for each maize plant. This topsoil was composed

of 72% dry matter and organic matter represented 5.9% of the dry matter content. Water holding capacity was estimated at 150% by the supplier, conductivity was 1.36 mS/cm and pH was equal to 7.5. The 60 plants were equally distributed between the five treatments: 12 Maize plant only (Maize), 12 Maize infested by *D. virgifera virgifera* (maize + WCR), and three treatments involving maize infested by *D. virgifera virgifera* and inoculated with *G. aculeifer* (maize + WCR + mites) with various mite densities: (i) 12 plants with 100 *G. aculeifer* (WCR + 100 mites), (ii) 12 plants with *D. virgifera virgifera* + 500 *G. aculeifer* (WCR + 500 mites) and, (iii) 12 plants with *D. virgifera virgifera* + 1000 *G. aculeifer* (WCR + 1000 mites). Pots placement within the greenhouse and the distribution of treatments over pots were both randomized to prevent any edge effects.

For treatments with WCR infestation: 21 eggs of *D. virgifera virgifera* sub-divided in three separated Eppendorf tubes were introduced at three distinct spots around each plant, at a depth of 10 cm beneath the soil surface [8] at sowing. For treatment with mites: A mix of *G. aculeifer* stages were then introduced at the estimated time of WCR larvae emergence (300-degree days base 11 after WCR eggs introduction, [44]) as previous work suggested that eggs are not identified as prey by *G. aculeifer* [37].

Maize development was assessed by monitoring the nitrogen levels of maize. Nitrogen deficiency is a commonly used indicator to assess WCR physiology impact on plants [55]. Nitrogen level was measured using a N-tester Nitrogen testing device commercialized by Yara [56]. This device optically measures the leaves chlorophyll content, which is correlated with plant nitrogen level. Thirty measurements are necessary to obtain one value for chlorophyll content per plant. These measurements were taken on the youngest fully developed leaf of each plant for each group when plants were at the 7–8 ligulate leaves stage. In order to assess the impact of the presence of various mite densities on *D. virgifera virgifera* development, WCR adult emergence was monitored. WCR adults were encased around each plant using tights as insect-proof nets. Traps were glued on the inside of each pot before adults emerged [44], and checked daily for emergence during 60 days. However, inspection of all parts of the pot was also necessary since adults could be found on the soil surface, on the plant or on the net. WCR adults were removed from pots and traps after each monitoring and placed in 70° alcohol to prevent spread in the trial area. Finally, to assess the potential damages induced by WCR larval stages on the root system of maize plants, each maize plant was pulled and washed out once adult emergence was over. The root system damage was ranked, using the widely used 0–3 IOWA ranking scale [57]. The lower level, i.e., the grade "0", was used when no visible damage was present; the grade "1" was used when damage marks were visible. The grade "1.5" was used when damage marks were visible with less than three roots severed and the higher grade i.e., grade "3", was used when damage marks were visible with three or more than three roots severed.

### 2.1.4. Statistical Analyses

All statistical analyses were carried out using R software (R Development Core Team, version 4.0.3). The chlorophyll index or the residuals of the following models involving chlorophyll index as the dependent variable followed a normal distribution, confirmed by a Shapiro–Wilk test and visual interpretation of quantile–quantile plots, and were therefore analyzed using Linear Models (LM). By contrast, the number of WRC emerged or the residuals of the following models involving the number of WRC emerged as the dependent variable did follow a Poisson distribution (commonly used to represent count data distribution) and were therefore analyzed using Generalized Linear Models (GLM). The root system damage estimated using the 1-3 IOWA ranking scale was analyzed using the Mood's median test enabling the comparison of ordinal dependent variables ('RVAide-Memoire' package).

To test the hypothesis that the presence of a predatory mite (i) reduces the population of WRC, (ii) limits their impact on plant nutritional quality (N level), and (iii) reduces the root system damages in the greenhouse trial, the impact of the treatments (maize vs. maize + WRC vs. maize + WCR + mites) on the number of emerged *D. virgifera virgifera* adults, the

chlorophyll index per plant and the 1–3 IOWA ranking scale were evaluated, respectively (treatments using varying mite densities were pooled in this analysis). The impact of the mites presence has been detected and in order to identify which density of predatory mite is necessary to control WRC populations with a limited impact on plant quality under the same conditions, we tested the impacts of the mite density (100, 500 or 1000 individuals per plant) in the maize + WCR + mites treatment on the number of WRC emerged, chlorophyll index per plant and the 1-3 IOWA ranking scale.

Multi-comparison tests were used to compute pairwise comparisons of the number of emerged *D. virgifera virgifera* adults and the chlorophyll index among the treatments (maize vs. maize + WRC vs. maize + WCR + mites) using the 'multcomp' package (Tukey method). Multi-comparison tests of the 1–3 IOWA ranking scale among the treatments were performed using the Bonferroni adjustment method.

### 2.2. Field Trial

### 2.2.1. Plant Material

For the field trial, the maize variety used was DKC5830, commercialized by "DEKALB" (Annexe 1). The seeds used were produced in 2020. Seeds were sowed the 17th April 2020 using a pneumatic plot drilling machine regardless of the treatments.

### 2.2.2. Arthropods Production and Preparation

*Diabrotica virgifera virgifera* were naturally present in the field of experimentation.

Sterilized *Aleuroglyphus ovatus* eggs were used as an alternative food source for predatory mites during the field trial. These sterilized *A. ovatus* eggs are produced by Bioline AgroSciences and commercialized as Predafix© Bioline AgroSciences, Clacton-on-Sea, England., They were formulated as aggregated particles, whose formulation is currently protected by a consortium agreement. These particles were produced in March 2020 and stored for 33 days at 4 °C before sending. After arrival at the research site, they were kept 44 days at 2.2 °C. Calibrated 0.25 g doses of formulated sterilized *A. ovatus* eggs (~126,400 eggs) were introduced in the entire maize plot immediately after sowing at a depth of 5 cm. A micro granulator was used to ensure homogeneous distribution of the product in the furrow.

*Gaeolaelaps aculeifer*. In the field trials, mites were produced by Biological Services, Australia (product and commercialized as Killer mites©). Predatory mites were stored for 71 days before introduction in climatic chambers at $25 \pm 0.5$ °C and $70 \pm 10\%$ RH with a constant obscurity on a substrate made of 1/3 blond sphagnum peat and 2/3 of fine (0.5 to 2,8 mm) vermiculite. A mixture of all *A. ovatus* stages was used as food, and extra water was provided three times a week in a $325 \times 150 \times 135$ mm$^3$ Bugdorm© Megaview science corporation, Taiwan with 30 μm nylon screen port sold by Saulas. Mites counting followed the same process as in the greenhouse experiment (see greenhouse trial) except the mites and substrates were prepared in 500 mL saltshakers. Due to the lack of knowledge about the potential effect of a mechanical introduction on mites' viability, mites were spread by hand on a micro-plot of 1 m$^2$.

### 2.2.3. Experimental Setup

The experiment was performed in an irrigated maize field located in Castagnole Piemonte, in Piedmont Region (Northwestern Italy) in 2020 (coordinates: 44.90603611 N; 7.57085277 E). Soil composition and environmental conditions are provided in Annex 1. This field was chosen because WCR was already present at high density and source of yield loss. No additional WCR individuals were introduced. The trial area was set up with at least 50 m of field edge to prevent any edge effects. An area of 35 m length (48 rows) per 10 m width was isolated from the surrounding field treatment by an additional 10 m width with no treatment zone as a buffer zone. Among the 48 rows, 12 plots of four consecutives rows were designed and treated. Within each plot, only the two central rows were sampled to prevent edge effects. Plots were assigned randomly to one of the treatments giving this

sequence: (i) 100 mites in each microplot over an area of 1 m$^2$ + 0.25 g/plant of alternative food composed of *A. ovatus* eggs particles ("Biological control treatment"), (ii) no pest control treatment ("control treatment") and (iii) FORCE Ultra pesticide application on the furrow during the sowing at 12.2 kg/ha ("pesticide treatment"). This sequence of biological control/control/pesticide was kept for the nine remaining plots. In each "biological control" plot, mites were spread on four micro-plots of 1 m$^2$ (c.f production of *G. aculeifer*), WCR population assessment and damage on maize were conducted only on this micro-plot whereas assessment for control and pesticide treatment were conducted randomly over the entire surface of each plot.

Maize development was assessed by monitoring the nitrogen levels of maize at two different plant development stages: when the plant had 7–8 leaves (12 June) and during the flowering period (16 July), for four plants randomly selected in each plot. Nitrogen levels were measured using a SPAD testing device commercialized SDEC. This device optically measures the chlorophyll content of leaves which is correlated with plant nitrogen level [58]. The thirty required measurements were taken on the youngest fully developed leaf of each plant for each group when plants were at the 7–8 ligulate leaves and at the flowering stage. In order to assess the impact of the presence of predatory mites (associated with alternative food source) on *D. virgifera virgifera* population dynamics, the number of emerging WCR adults was assessed. In each plot, 4 emergence cages were set up before first emergence. One plant is trapped inside a metallic cylinder of a diameter 25 cm and 60 cm high which half of it is inserted in the soil. A net is attached to the cylinder and maintain taut with a stick to prevent the net to impact maize growth. This device helps avoid the escape of WCR adults. Individual removal from the net is done through a zipper present all along the net length. WCR adult collection started once first emergence was recorded on yellow chromotropic trap placed in the field, then after 0, 8,16, 22, 29, 37 and 45 days, WCR adults were counted and removed from the cage. Finally, the assessment of the potential damages induced by WCR larval stages on the root system of maize plants was realized as described above. Four maize plants per plot were pulled on 16th July and washed out once adult emergence was over. The root system damage was ranked, using the 0–3 IOWA ranking scale [57].

### 2.2.4. Statistical Analysis

All statistical analyses were carried out using R software (R Development Core Team, version 4.0.3). The chlorophyll index or the residuals of the following models involving chlorophyll index as the dependent variable followed a normal distribution, confirmed by a Shapiro–Wilk test and visual interpretation of quantile–quantile plots, and were therefore analyzed using Linear Mixed Models (LMM). By contrast, the number of WRC emerged or the residuals of the following models involving the number of WRC emerged as the dependent variable followed a Poisson distribution (commonly used to represent count data distribution) and were therefore analyzed using Generalized Linear Mixed Models (GLMM). The root system damage estimated using the 1–3 IOWA ranking scale was analyzed using the Mood's median test enabling the comparison of ordinal dependent variables ('RVAideMemoire' package).

In the models analyzing the number of emerged *D. virgifera virgifera* adults per plant, the chlorophyll index in both the 7–8 leaves stage and flowering stage and the 1–3 IOWA ranking index, pest control treatment (no treatment vs. introduction of both mites and food vs. pesticide application) was modelled as a fixed effect. In the LMM, each plant represented a technical repetition (30 and 20 measures have been taken for each 7–8 leaves stage and flowering plant, respectively) and was modelled as a random effect to consider the dependence of replicates from the same plant. In the GLMM, the emergence cage was modelled as a random effect to consider the dependence of replicates from the same cage.

Multi-comparison tests were performed to compare the number of emerged *D. virgifera virgifera* adults per plant and the chlorophyll index among the pest control treatments (no treatment vs. introduction of both mites and food vs. pesticide application) using the

'multcomp' package (Tukey method). Multi-comparison tests of the 1–3 IOWA ranking scale among the treatments were performed using the Bonferroni adjustment method.

## 3. Results

### 3.1. Greenhouse Trials

The chlorophyll index varied depending on the treatments (maize vs. maize + WRC vs. maize + WCR + mites) but not the mite density in the maize + WCR + mites treatment (Table 1, Figure 1). Using pairwise comparison, the control mean chlorophyll index (590 ± 7.04) was 19% higher than the maize + WCR treatment (484 ± 16.9, $p < 0.001$) (Figure 1). The maize + WCR + mites treatment had a similar Chlorophyll Index (586 ± 7.04) as the control treatment ($p = 0.926$), and significantly higher than the WCR-only treatment ($p < 0.001$).

**Table 1.** Results of Linear (F value) and Generalized Linear Model ($\chi^2$ deviance) on the number of adult *D. virgifera virgifera* emerged per plant, the nitrogen content of plant, and the root damage index (IOWA) in both greenhouse and field. Factors included in the models are described as explanatory variables. Significant effects are indicated in bold text. Df correspond to the degree of freedom.

| | Variables to Explain | Number of Adult Emerged | | | Chlorophyll Index | | | | | | Root Damage Index | | |
| | | | | | Young Plant | | | Flowering Plant | | | | | |
| | Explanatory Variables | Deviance | Df | *p* Value | F Value | Df | *p* Value | F Value | Df | *p* Value | Deviance | Df | *p* Value |
|---|---|---|---|---|---|---|---|---|---|---|---|---|---|
| **Greenhouse Experiment** | Maize vs. Maize + WCR vs. Maize + WCR + Mites | 151.0 | 2 | **<0.001** | 34.6 | 2.56 | **<0.001** | | | | 23 | 2 | **<0.001** |
| | Maize + WCR + 100 Mites vs. Maize+ WCR + 500 Mites vs. Maize + WCR + 1000 Mites | 1.3 | 2 | 0.532 | 0.13 | 2.33 | 0.877 | | | | 2.9 | 2 | 0.235 |
| **Field Experiment** | Control vs. Biological control vs. pesticide | 67.1 | 2 | **<0.001** | 23.9 | 2 | **<0.001** | 1.5 | 2 | 0.223 | 56.4 | 2 | **<0.001** |

Root damages represented by the IOWA index varied significantly according to the treatments but not the mite density in the maize + WCR + mites treatment (Table 1, Figure 1). The mean IOWA Index for maize + WCR treatment was close to the highest possible value (2.55 ± 0.24) and significantly higher than both the maize (no damage) and the maize + WCR + mites treatment (0.57 ± 0.12; $p < 0.001$).

The number of emerging adults also varied depending on the treatments but not the mite density in the treatment using *G. aculeifer* (Table 1, Figure 1). The number of adults was close or equal to zero in control pots and in maize + WCR + mites (0.56 ± 0.12) treatment while on average 6.1 ± 0.48 adults per plant were captured in maize + WCR treatment ($p < 0.001$).

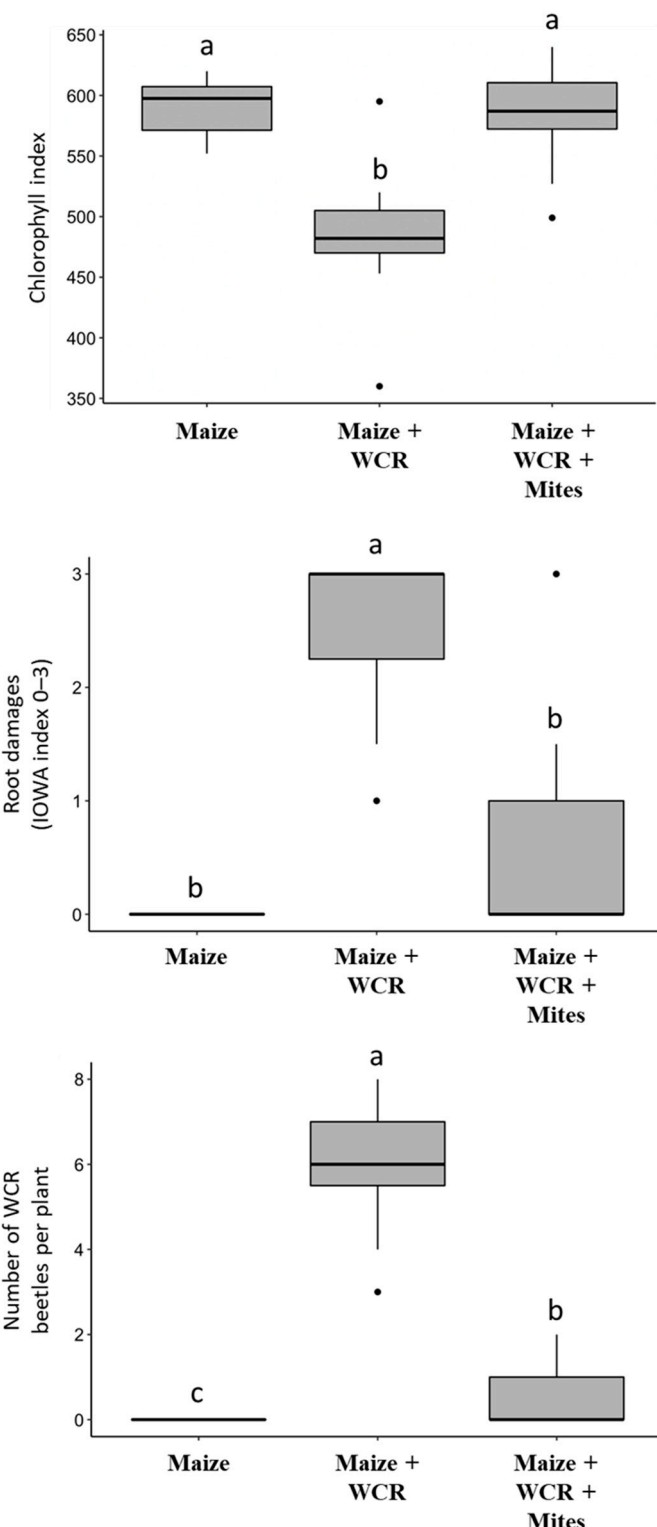

**Figure 1.** Greenhouse trial. Impact of the maize treatments (maize vs. maize + WRC vs. maize + WCR + mites) on the chlorophyll index (N-tester; top panel), the root-damage index (IOWA index: 0–3; middle panel) and the total number of adults emerging. Boxplot topped by the same letters did not differ significantly.

### 3.2. Field Trials

The SPAD reading showed slight, but significant differences among the three treatments for the 7–8 ligulate leaves stage (control: 36.6 ± 0.38, pesticide: 38.81 ± 0.49) (Table 1,

Figure 2) with an advantage for the biological control treatment ($41.04 \pm 0.48$; all $p \leq 0.001$). These differences did not persist at the flowering stage (control: $48.43 \pm 1.03$; pesticide: $50.63 \pm 0.93$; biological control: $50.08 \pm 0.85$; Table 1).

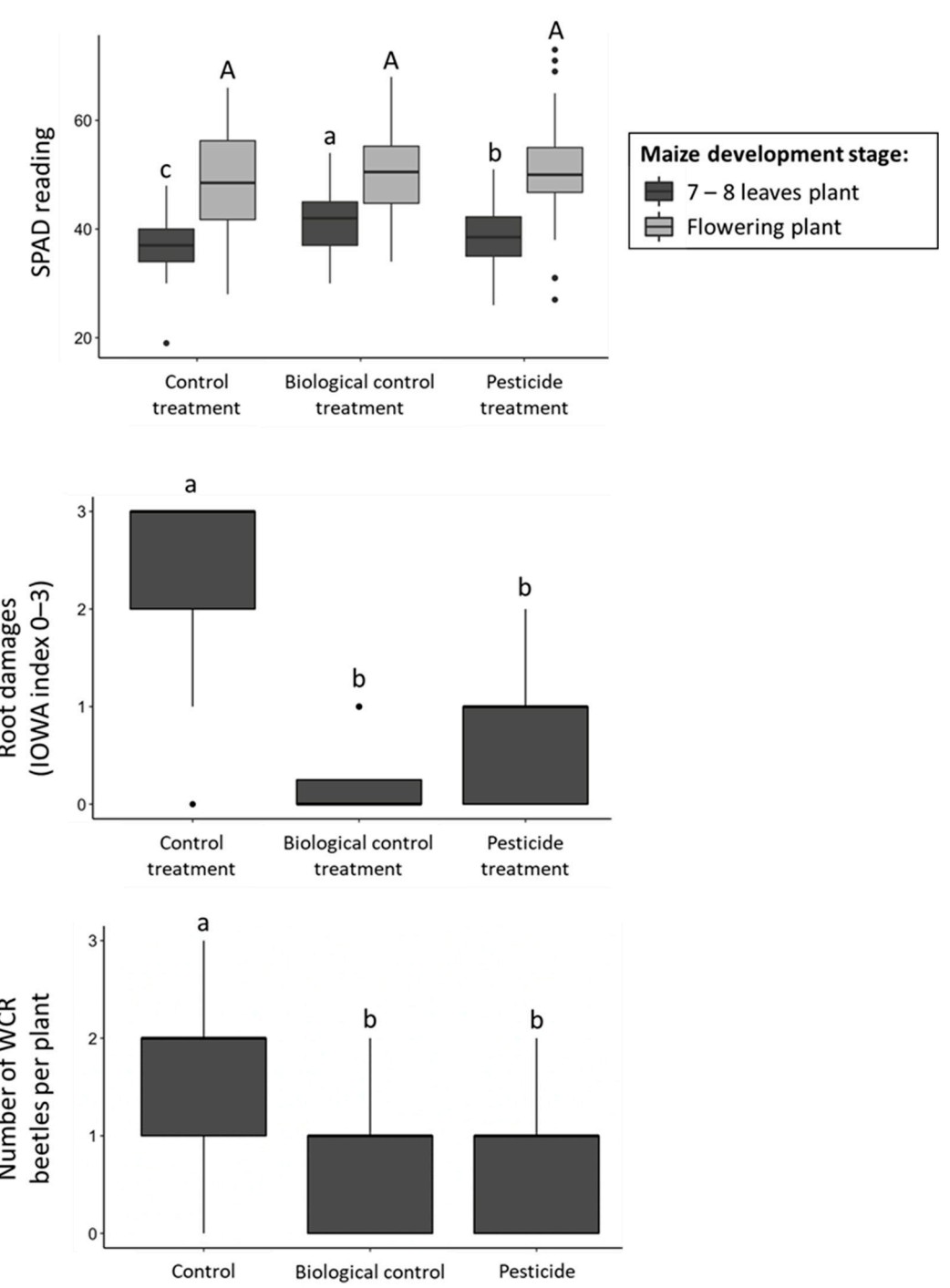

**Figure 2.** Field trial. Impact of the treatment type (control, biological control through the use of predatory mites and pesticide treatments) on the maize chlorophyll index (SPAD-reading; **top** panel), the root-damage index (IOWA index: 0–3; **middle** panel) and total number of WCR adults emerging per plant (**low** panel). Boxplot topped by the same letters did not differ significantly.

Root damages represented by the IOWA index also showed a significant and similar impact of both pesticide ($0.73 \pm 0.10$) and biological control ($0.25 \pm 0.11$) treatments when



compared to the control ($2.45 \pm 0.13$; all $p < 0.001$) (Table 1, Figure 2). Despite very little variation and the lowest mean response, results from the biological control treatment were not significantly different from those obtained under the pesticide treatment ($p = 0.46$).

WCR adult emergence, represented by the mean number of WCR per plant, varied significantly with the treatments (Table 1, Figure 2) and was higher in the control treatment ($1.52 \pm 0.09$) than either in the pesticide ($0.65 \pm 0.06$; $p < 0.001$) or the biological control treatment ($0.54 \pm 0.05$; $p < 0.001$). No significant differences among biological control and pesticide treatment were found ($p = 0.50$).

## 4. Discussion

Our results provide the first empirical demonstration of the ability of soil-dwelling predatory mites to prevent WCR damages on maize plants. Furthermore, we show preliminary evidence that a predator-in-first strategy may provide a level of control equivalent to pesticide treatment, while reducing the costs related to the production and introduction of predatory mites.

To evaluate WCR impact on plants and approximate potential yield loss, we used indicators of plant health, WCR population and WCR damages. We first estimated larvae density and development by observing root damage using a 0–3 scale called IOWA index, a proven reliable indicator of WCR damage on roots. However, observable damage does not necessarily induce physiological deficit for the plant due to plant resilience to pest attack [59]. We thus collected the chlorophyll Index in order to identify the global physiological impact of WCR on the plant once larvae already started to attack roots (7–8 leaf stages). Then, we assessed adult emergence to verify whether the biocontrol treatment affects the capacity of WCR to achieve its life cycle. We showed that adult emergence, root damage and chlorophyll index were all correlated indicators of the WCR population impact on maize plants. These parameters can thus be used confidently to assess the capacity of treatments to control WCR population and limit their impact on maize plants.

We considered that treatments in both trials were conducted on surfaces insufficient to have a representative yield loss, but Haegele et al. [60] highlighted the strong relationship between chlorophyll and grain yield for maize. Urías-López et al. [55] showed that decreases of the chlorophyll index in maize attacked by WCR led to a reduction in plant height. Many studies showed that WCR physiological impact on the plant can lead to significant yield loss [59,61–63]. In both experiments (greenhouse and field trials), chlorophyll index in young plants decreased in the presence of WCR larvae and remained high when pest populations were absent or controlled (by pesticide treatments or using predatory mites). These results confirmed that decreasing WCR population density would improve plant physiology and yield. Previous work [64] confirmed this statement by showing that four adults per plant induced damage on maize, whereas one adult per plant did not decrease yield. The WCR density for biological control and pesticide treatment was below one in our study even if this threshold has to be used carefully since plant response to WCR attack also depends on plant health [62]. Conversely, the presence of pests does not seem to modulate the chlorophyll index when the plants have reached the flowering stage. However, according to Urías-López et al. [55] plant height or grain yield could already have been impacted by chlorophyll index decrease at younger stages.

All indicators showed that the inundative strategy allowed us to control WCR population and preserve plant health. The greenhouse experiment aimed at determining the optimal mite density that should be introduced to achieve efficient control of WCR larvae in a soil. The three mite densities tested (100, 500, or 1000 mites per plant) showed similar and positive results since both WCR population density and root-damage index decreased significantly, while physiological plant indicator (chlorophyll index) improved with mite presence. Further studies were planned to assess effect of *G. aculeifer* smaller density on WCR population, but they were not feasible within the timeline of the project due to global conditions (2020 covid-19 pandemic). Minimum *G. aculeifer* density threshold to control the WCR population at the larval stage remains to be determined.

*Gaeolaelaps aculeifer* timing release is a key element for the successful control of WCR population. A previous study conducted by [65] also tested the *G. aculeifer* inundative strategy against WCR. They used density comparable to what we introduced in the greenhouse trial (150 mites/plant or 300 mites per plant) and surprisingly showed no effect on WCR population. The main difference between their trial and ours is the timing of the introduction of the predatory mites. In their study, they introduced mites and WCR eggs at the same time whereas we introduced mites secondarily at the estimated time of larvae emergence. Soil-dwelling predatory mites and especially *G. aculeifer* showed a very high capacity to disperse to find food and sufficient humidity (personal observation). We assume that in the Prischmann-Voldseth and Dashiell experiment, *G. aculeifer* might have spread away from the plot while the eggs were maturing, so that they were not present around the plant when the eggs hatched. Alternatively, if the amount of natural resources in the soil was insufficient to sustain the mite population, the number of mites surviving until WCR larvae emergence might have been greatly reduced. This highlights the need to reliably sustain the population of predatory mites close to the plants over a prolonged period until WCR larvae emerge.

On the field scale, 100 mites per plant still represent a large number of, but the alternative predator-in-first strategy we tested in the second experiment allows reducing the density introduced while maintaining efficiency. In the greenhouse trial, 100 mites per plant were introduced. Since a maize field contains up to 90,000 plants per hectare over extended surfaces [1], this quantity is not expected to be viable in agronomic conditions. To reduce the density, 100 mites/m$^2$ with alternative food were introduced in the field trial, amounting for approximately 11 mites per plant. The density was then expected to reach higher levels at the time of WCR larvae emergence and keep them along the furrow. We found that this predator-in-first strategy coupled with the introduction of alternative food was efficient to control WCR populations, with a performance equivalent to the pesticide treatment. The introduction of a low number of individuals at sowing with alternative food has many agronomic advantages. It could decrease the purchase cost of predatory mites, the production of these organisms being particularly expensive with current technology. The predator-in-first strategy is expected to reduce the workload of farmers by allowing them to introduce the individuals at the same time as planting. It also helps to ensure that the predatory mites are present in the soil at the vulnerable stage of WCR development, as the precise timing of WCR larvae emergence in natural conditions can be difficult to predict [45]. The predator-in-first strategy was first developed on *phytoseiid* predatory mites to control thrips [46]. This method's main concept is to introduce a predatory population before the peak of pest emergence, so that the predator population is established and amplified when the sensitive stage for predation occurs. The main limit to this strategy is that it requires predators to be able to feed on alternative food sources, but in our case *G. aculeifer* showed its capacity to feed on many different food sources such as pollen [66], dust mites *Tyrophagus putrescentiae* [67] or eggs from *Aleuroglyphus ovatus* [54]. The latter were used in our experiment with a small amount of predatory mites introduced at sowing in order to initiate the population. Our results showed that predatory mites, with alternative food, were highly efficient protecting the plant against WCR under field conditions. Indeed, all the indicators used in the experiment lowered pest pressure on roots, decreased adult emergence and increased chlorophyll index at 7–8 leaves compared to the area without treatment. Moreover, biological control showed an efficiency comparable to pesticide treatment, enhancing its potential as an alternative to the common solution in this pest management. More than a promising result for WCR control, it showed that biological control relying on population dynamics and the predator-in-first strategy can help farmers in their work. Indeed, the predator-in-first strategy decreases the necessary amount of biocontrol agents that need to be introduced and thus reduces cost. It does not require knowledge of larvae emergence timing to spread solution, and the farmer may introduce the biological agent simultaneously with another mechanical intervention.

Natural predation is unlikely to have caused WCR population control in our field trial. In the present study, we added *Aleuroglyphus ovatus* eggs in the soil to enhance *G. aculeifer* population. The introduction of an alternative food source could by itself attract other predators such as other predatory mites species or *Staphylinidae* also present at the top of the food chain in crops [31]. However, [68,69] highlighted the negative impact of continuous crops and pesticide on predatory organisms. Since both these factors are frequently present in maize crops, we assume that natural predatory density is degraded enough to prevent it from controlling the WCR population in most agronomical contexts. This assumption is confirmed by the lack of natural control by endemic predators both in North America and in Europe and the necessity to apply culture rotation to lower WCR population density [14] despite years of WCR presence in the soil. Our own samples from three maize fields in three maize production areas in France and Italy also showed extremely low natural densities of soil predators (T. Andrieux, personal communication, less than 200 predatory mites for 650 samplings of approximatively 150 g of wet dirt).

The potential long-term presence of predatory mites in soil has not been considered yet. *G. aculeifer* presence in the soil and its capacity to overwinter suggest that temperature conditions are not the issue for the population to maintain itself from one spring season to another [43]. The second essential parameter, a high level of humidity in soil, is not considered as a limiting factor either because a low density of individuals is still present in the soil meaning they found appropriate conditions to survive all year long. We assume that food availability is the most likely limiting factor for sustaining predatory mite populations at sufficient levels to control WCR, because of low biomass in continuous maize fields [68,69] (e.g., 1,94% organic matter in the Castagnole Piemonte field see Supplementary Materials). We assume the PIF strategy we explored in this work can help to transition from a chemical pesticide habit which significantly impacts biodiversity and biomass to a more integrated management by providing supplementary biomass in this damaged environment.

Adopting relevant farming practices could enhance predatory mites efficiency to control WCR. Kautz et al., (2006) highlight the positive impact of different types of manure introduction on predatory mite populations, whereas Cortet et al., (2002) showed a negative impact when tillage is applied. Hamers and Krogh, (1997) concluded the negative effect of pesticides on *G. aculeifer* specifically. In the scope of reducing the impact of agronomic practices on the environment, these conclusions suggest that adopting European legislation recommendation (directive 2009/128/EC) could further improve the performance of the biocontrol solution we investigate in this study.

As a generalist predator, *G. aculeifer* could be used against another major maize pest. Wireworms are another coleopteran species able to feed on maize roots and many other crops of economic interest [70]. In previous work in laboratory conditions ([37], *G. aculeifer* showed its capacity to attack *Agriotes sordidus* first instar larvae, but these promising results will require further investigation in field conditions. *A. sordidus* neonate stages are asynchronized with WCR neonate emergence [8,71] which would complicate a single, inundative introduction strategy. In contrast, a strategy based on the early introduction of soil-dwelling predatory mites combined with alternative food, as we proposed in the present work could maintain the population through both emergences to control both population pests.

This work is definitely promising but requires further investigation to confirm its potential. As we mentioned earlier, the population dynamics of predatory mites, and biological control agents in general, highly depend on environmental conditions such as food availability, temperature and humidity [72,73] which in turn depend on agronomical practice such as pesticide use, irrigation, manure or tillage [74,75]. The predators-in-first strategy lets the biocontrol agent confront these conditions for a long period and we only tested this method in one scenario of predatory mite dose, WCR density, food source quantity, temperature, humidity, soil composition, whereas each of these parameters can impact positively or negatively on mite's efficiency to control the WCR population. This

work will have to be particularly addressed to consider an applicable biological control solution against WCR in all its areas of nuisance [76].

**Supplementary Materials:** The following are available online at https://www.mdpi.com/article/10.3390/agronomy11101984/s1, Field Trial Report: Efficacy and crop selectivity of *Aeuroglyphus ovatus* egg's strategy against *Diabrotica* spp. (DIABVI) on Zea mays.

**Author Contributions:** A.P., T.A., M.F., A.M., B.K., C.C. and E.V. conceived and designed the experiments; A.P., T.A., A.M., B.K. and C.C. performed the experiments; A.P., L.S.M., M.F. and E.V. analyzed the data; and A.P., L.S.M., M.F. and E.V. wrote the paper. All authors have read and agreed to the published version of the manuscript.

**Funding:** This research was funded by SEMAE (previously The French Interprofessional Organization for Seeds and Plants (GNIS)) within the Mites Against Diabrotica project.

**Institutional Review Board Statement:** Not applicable.

**Informed Consent Statement:** Not applicable.

**Acknowledgments:** We are thankful to the French Interprofessional Organization for Seeds and Plants (GNIS, newly SEMAE) which enabled this scientific work by funding the Mites Against Diabrotica (M.A.D.) project. We also acknowledge all the partners in this project: Axereal Serbia, French National Institute for Agricultural and Environmental Research (INRAE) and Arvalis.

**Conflicts of Interest:** The project was partly funded by the company BioLine agrosciences, who develops and commercializes biocontrol products.

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
