# Peer review of "A Promising Predator-In-First Strategy to Control Western Corn Rootworm Population in Maize Fields"

_agronomy, doi:10.3390/agronomy11101984_

Round 1
Reviewer 1 Report
This is a potentially important study looking at a very poorly studied topic: the role of soil predators to control soil pests. I recommend it for publication but after a very major revision since there are several important constraints. All the sections require a revision, particularly M&M section that needs significant improvements.
Remarks and suggestions in details are reported below.
Abstract: the abstract has to contain a brief description of M&M applied; currently there is nothing; please insert experimental semi-controlled and field methods; ideally, all the main paper sections should be pinpointed very briefly in the abstract (problem description/background; definition of the goal/goals; material and methods, results and conclusions)
Introduction: The first part is misleading and needs a literature update; I can understand that the authors try to outline the importance of their work but they cannot hide the updated information giving a wide description of pest status; if you cite Wesseler and Fall, 2010 [9], you should at least cite Bazok et al. 2021 https://doi.org/10.3390/ reporting that the scenario described in 2010 did not become reality and that in Europe rotation implementation allowed a concrete satisfactory WCR management without a catastrophy; likewise, the threat rising from Curzi et al. 2012 [14], has to be considered but the current situation is that described in Bazok et al. 2021 https://doi.org/10.3390/: rotation works well in Europe; please reword all the first part, balancing information available; IPM principles and European legislation (namely Annex III of directive 2009/128/EC) should be considered for this discussion as well;
L 81: you have to give the name before the citation number; this unpleasant citation system does not exclude that you can mention authors before the number when the sentence requires this; on the contrary this impoves text readability;
Information about the mite predators should be given; are there naturally present in European fields? What about their biology? Which are the most suitable conditions for their development? Etc....
What about the interactions between the generalist mite predators and other soil dwelling species?
How can I manage the mite predators in a holistic integrated cultivation and pest management?
Materials and Methods
Statistical methods are part of M&M, numbers have to be changed
They are poor: essential information needed to evaluate and exploit the work is hidden. A complete accurate description of methods should be given. Any method you use in a scientific research has to be objective, effective and replicable by anyone; therefore all the details needed to evaluate and replicate the experiment have to be given. M&M should start from operational protocols you set up to plan and implement the trials; this would allow any reader to replicate properly the experiment;
Field trials
Site description – please give geographical coordinates of field experiment site;
Real soil characteristics (% of silt, clay, sand, organic matter (C) of the fields under experiment should be given);
Climatic conditions during experiment
Agronomic information (tillage, sowing date, pesticides applied – herbicides, insecticides,…with timing) and climatic parameters (T, soil moisture,……)
L 248-249: 25cm*25cm square of what? Which type of cage did you use? Which materials was it made of? How did you search the bbetles making sure they could not .escape at inspections? Etc.
An evaluation of the presence of other harmful organisms and abiotic factors that might have interacted with crop status should be given; an estimation of soil biodiversity (QBSar cna give an estimation of soil biodiversity) to have a representation of teh actual ecosystem the generalist predators dwelled.
Results
In order to get solid results, this kind of experiments should include at least two field seasons.
Discussion
The results should be exploited in a holistic approach; considering all the potential factors playing a role, including interactions with other soil beneficials and predators. We have to deal with ecosystems made up of many components that continuously vary genetically and interact with one another; all of our intervention (e.g. tillage, fertilization, irrigation, pesticide application or predator application,….. triggers new interactions between ecosystem components. You should discuss how the introduction of a generalist predator may have side effects; might generalist predators populations decrease other beneficial populations and make possible the outbreaks of other soil pests (e.g. you can control WCR larve but you may cause the outbreak of other maize soil pests due to the killing of their antagonists by the introduced predators? Another possible scenario might be the opposite: introduced generalist predators may kill other soil pests causing a complete crop protection. What can the authors say?
L 428: This key statement is unjustified; the authors should give references, or at least an organic demonstration to justify their statement; again, an holistic approach implementing the agronomic factors that increase soil biodiversity might create conditions to allow soil benficials to reach population levels suitable for control WCR and other soil pests at the desired level. The more complex the ecosystem, the more stable it is.
L 433: continuous maize not monoculture
Another constraint of the manuscript is that it describes short term experiments; key questions remain open: e.g. were mite predators active in the subsequent season?
Your approach was to use beneficials (generalist predator mites) the same way chemical insecticides are generally used; this is a welcome potential alternative to chemical insecticides but an integrated approach creating soil conditions for long term presence of beneficials (increasing of biodiversity) would make more sustainable the predator use.
Author Response
Please find attached the response to reviewer 1

Reviewer 2 Report
A promising Predator-In-First strategy to control Western Corn Rootworm population in maize field.
Pasquier, et al.
Comments about the manuscript ID: agronomy-1343580
- INTRODUCTION
According the title, this work approaches a promising predator-in first strategy to control WCR in maize field, making it very clear what the objective of the work is.
However, more than half of the introduction draw attention to different WCR control strategies (mechanical intervention, with Hymenoptera, Diptera, nematodes, fungi), which are not relevant in this context, without focusing on the subject showed in the title. Only the last ten lines (101-110) refer to the strategy "predator-in first" without providing a strong work hypothesis and comprehensive aim of study.
The aim of the study is unclear (lines 89-101).
First, the authors aim to “test a potential of G. aculeifer to control WCR in more realistic conditions. ......” releasing predatory mites around plants infested (lines 89-94). However, it is no clear that this objective is going to carry out in this study by application limitations (lines 96-100).
Later, "based on all these constraints” (better - by all these constraints) of the realistic method, the authors “explore another strategy in a field trial, the Predator in First” (lines 96-110).
After reading this part of the introduction and as it is written, it is not known whether the authors are going to: 1) address a study about the effectivity of G. aculeifer to WCR larvae in the soil (assessing the limitations of this method); 2) address the PIF strategy and / or; 3) address both strategies to compare them each other.
- MATERIAL AND METHODS
The material and methods are not adequately described, which make this epigraph incomprehensible.
In this paper, the years of experimental work and seasons are not specified. In addition, to know the lengths of time in which the experiments are maintained is necessary because the time can be an important factor in the treatment success.
In the lines 117, 148, 282 and 283 is mentioned in-situ trials, which are not specified in objectives and it is not known what it is. The name of the experimental work should be homogenized (if in-situ trials = field trial).
Arthropods production
The authors specify an epigraph Arthropods production, but Institutions (Hungary´s CABI and EWH bioproduction) provide the insects and mites. Presumably, the authors maintain the mite population in vermiculite for 8 months at 70% RH and in dark, but it is surprising that the rearing be successful without fungal attacks for so long period of time.
Alternative food
It is not comprehensible that the authors include the experiment descriptions into the epigraph Alternative food (Experiment 1: semi-controlled conditions and Experiment 2: field trial)
According to the objectives, two strategies are being assessed:
- a) To release predatory mites around single maize plants infested with WCR in semi-controlled conditions (I guess in greenhouse).
- b) Predator-in-First in field trial (lines 101-102)
The experimental design is not clearly specified under these two strategies. The description (lines 166-254) include different concepts exposed without order, which introduce chaos in the explanation. All these concepts should have separated in epigraphs to clarify the methodology.
- a) Greenhouse experiment
- The climatic conditions are specified without order in different paragraphs (lines 168-172 and lines 190-195). The control of Tº and RH is not clear when the Tº is reduced (how much?) and increased the RH (how much?) (lines 171-172). The control of soil moisture is not accurate by watering with one liter or two of water (lines 191-195).
- The design of experiment is confusing and without order.
It seems that the surface used for the experiment is 48 m2 (~ 7 m x 7 m) with 6 rows with 10 maize plants/row and 5 treatments: 1) Maize (only); 2) maize + WCR; 3) maize + WCR + 100 mites; 4) maize + WCR + 500 mites; 5) maize + WCR + 1000 mites.
If there are five treatment, each in one row (five row), then there is an extra row that is not known what it is used for. In addition, these five treatments are replicated 12 times each, but the surface of 48 m2 is very small for that. Are the authors using 48 m2 by replicate?
According to the authors, sixty pots were set up and equally distributed between the five treatments. They do not specify how is that distribution (here a schematic figure would have clarified a lot)
The authors do not explain why they introduce a mix of G. aculeifer stages at the time of WCR larvae emergence (300-degree days base 11; [41]) (lines 189-190). The quote [41] does not explain this decision.
Maize development was assessed by monitoring the nitrogen and chlorophyll levels of plants. The authors use the nitrogen deficiency as an indicator to assess the WCR influence on plants (lines 196 – 205). However, this device has problems to make this measures because the experiment has not enough plants. Then, why does it use?
“To overcome this limitation” the authors took the thirty required measurements on the youngest fully developed leaf of each plant (12) for each group. However, according to the experiment explanation the plants are 10/row (line 174) and the replicates 12 (line 175). It is neither comprehensible how many leaves they are using nor which groups are they referring.
- b) Field trial
The design of experiment is not understandable.
The authors are going to explore the strategy PIF in a field trial, (lines 101-110). According to the experiment 2 (field trial) this strategy is performed in a maize field chosen because WCR was already present at high density. However, here it is not possible carry out the strategy PIF, because the pest (WCR) is present at high density before releasing the predator (mite).
In the treatment of biological control, 100 mites (G. aculeifer?) are released in each micro-plot over an area of 1 m2 + 0.25 g /plant of alternative food composed of A. ovatus. It is not understand why A. ovatus is release because this mite could compete with WCR and to introduce a bias in the experiment.
The damages are assessed by nitrogen level using a SPAD testing device (line 242-243). Is SPAD device different to the N-tester Nitrogen testing used in greenhouse experiment? If it is, why do not the authors use the same Nitrogen testing for both experiments? In the figure 1, the authors show the locations of the N-tester not SPAD testing device in all plots? This is confusing.
As in the previous experiment (greenhouse), the authors do not explain correctly the number of plant used to get the nitrogen and chlorophyll measurements (lines 243-245).
The authors set up four emergence cages of 25cm*25cm to assess the impact of treatment 1- biological control (lines 245-249). Why do not they assess the area of 1m2 established for the treatment of biological control?
In addition, as the text is written, it seems that this emergence cages are installed at plots of biological control only, but the figure 1 shows that the cages are in all plots. This point is not clear in the text.
In order to assess the effectiveness of the treatments, the density of the WCR must be determined prior to the experiment.
Statistical analysis must be included in the epigraph of Material and Methods.
The authors do not explain in an understandable way what the statistical analysis purposes is to confirm or reject the working hypothesis (which is unknown).
In semi-controlled conditions, the authors do not explain why they carry out two GLM with the mite density (100, 500 and 1000) pooled and other GLM broken down. It is not clear whether LM is used with the mite density pooled or broken down and why LM is used.
In the field trial, the authors do not explain why they carry out two GLMM and why the emergence cage and the treatment are modelled as random and fixed effect respectively. Similarly, for LMM on the nitrogen levels in leaves and flowering stage (random effect).
In the statistical analysis, the mite’s successful addition in the plots with biological treatment has not been verified. Mites’ wrong addition in plots with this kind of treatment could lead to erroneous results.
The table 1 has to be specified in Results not in Material and Methods.
- RESULTS
The results are not clear and consistent enough. Taking into account the chaotic exposition of objectives and material and methods, it is not possible evaluate whether the results are adequate,
Semi-controlled conditions trials
The chlorophyll index is correlated with plant nitrogen level. The authors have never explained this correlation statistically. This is necessary to understand the results.
The Table 1 does not show the statistical results for the variable chlorophyll index. Neither, it is specified what model (GLM, LM, GLMM, LMM) is working with and whether chlorophyll index data met the assumption of parametric or not-parametric statistic.
Root damages and number of emerging adults. The values presented in the Table 1 does not agree with those of the text. There is more confusion in the results.
The Table 1 shows only two explanatory variables: Treatments and Mite density (Maize + WCR + mites treatment). However, in Material and Methods, the authors proposed five treatments: Maize (only); Maize + WCR; Maize + WCR + mites (100, 500, 1000). Where are the statistical analysis of all of them?
Field trial
As in the previous case, Table 1 shows only Pest control treatment. Where are the statistical analysis of all treatments?
The effect of releasing A. ovatus is not mentioned in any other epigraph. Has it been competitive with WCR?
Has the density of the WCR been determined prior to the experiment? How has it affected to the results?
Discussion
According to the authors, they show that a PIF strategy provide control in the field of WCR. However, in this paper, the strategy PIF is missing; the control strategy with three different mite densities is not specified in the results.
In conclusion, in this paper the authors, at most, have assessed the effects of releasing a generalist predators (G. acuelifer), which is not at all original because it has already been done before and this is not what the title promising.

Author Response
Please find attached the repsonse to reviewer 2
Reviewer 3 Report
Aug 16, 2021
Agronomy article review
A promising predotor-in-first strategy to control wesnter corn rootworm populations in maize field
45-74 This paragraph could be eliminated or summarized with two sentences that cites some of the efforts on biological control of WCR and give references.
Over the manuscript, the field and greenhouse studies are called semi-controlled and in-situ, experiment 1 and experiment 2. I suggest using “Greenhouse study” and “Field study” throughout the manuscript.
109: Last sentence can be eliminated, is part of the M&M
115-116: eliminate “One out of 60 maize plants…” sentence.
146: Was the content of the tubes just spread over the soil surface on the pots? Could the mites escape the pot?
149: Mites were stored for 71 days. Was the media used for storing them the same as for the greenhouse study?
153: “only small surfaces were treated”. This probably means that the introduction was done in an area of 1 meter square only, instead of the whole surface of the plots. I suggest not using the term “treated” when referring to the mites.
154: Was the process to count mites the same as in the greenhouse study? If not, how were the mites counted?
157: this paragraph refers to the field study, but is followed by the description of the greenhouse study. This makes it confusing.
The manuscript can benefit from simplifiying the M&M section. For example, details that are not critical should be eliminated (how the greenhouse was cooled, water holding capacity of substrate, how plants were watered, etc).
196: Authors should start by describing that the N tester is a standard tool to monitor WCR impact on plants, and then describe what they used. It is confusing to read about N testing without first knowing why.
201: It is not necessary to give the description of how it is done in the field, this is a greenhouse experiment. Authors should just describe how they used the N tester.
235: The description indicates that there was one, 1m2 area where the mites were released, but the graph shows four.
235: How was it ensured that the mites did not move further than the 1 m2 micro plot area? Are there some studies that show what is the scale of movement of the mites once they are released in the soil?
Fig. 1. The description of the study set up is clear in the M&M, I suggest eliminating figure 1.
266: why were mite densities pooled? Why not just analyze the five treatments?
268: Reference table 1 in the results, not M&M
261: this section is a little confusing. Describe the analysis per study or variable, but not both.
302: no need to give the p values in the text, they are in table 1.
Fig. 3. The mean separation for Spad reading, Pesticide treatment, it is probably a “b” or “a”.
341: the study shows that the predators have the capacity to lower the pest density, when introduced with alternative prey. Based on one study, it is not appropriate to conclude that the predatory mites are “equivalent to pesticide treatment”.
344: This paragraph can be substantially shortened. It is repeating what has already been described in the M&M
357: Since this study quantified WCR emerged, are there estimates of the effect on yield per WCR recovered. While quantifying the effect of WCR on yield was not the objective of this studies, it would help convey that the mites can reduce populations below levels of economic damage.
The authors bring an interesting point in paragraph 428. For the field study to dismiss some of the issues brought up, a treatment that included predatory mites without alternate pray and another treatment of only alternate pray would have been necessary. Their explanation of why other predators may not have influenced the results of the field study is not adequate. In my opinion, it would be better to mention that other predators may have played a role and that other studies would be needed to rule out that factor.
Overall, this study presents evidence that using predatory mites may constitute an alternative for managing of WCR (this is shown in both the greenhouse and field studies). The authors should avoid putting too much emphasis on the density of predatory mites needed since this is the result of only one study (the greenhouse study). Also, the authors should not generalize too much about the use of alternative prey since they only did one study (the field study).
Author Response
Please find attached the response to reviewer 3

Round 2
Reviewer 1 Report
The authors have properly addressed most the remarks but some shortcomings have remained.
Point 1: What does it mean aboveground maize plants? I suppose that any maize plant has an above ground part; you state that you used WCR eggs to infest the plants, it seems you did not infest the above ground part; or you released WCR adults to get egg laying?
Please reword the sentence;
Point 2: Before publishing all of us should search for all the last papers dealing with the topic; some papers to be considered are open access and very easy to find and download, then….
Point 6: Because the natural diversity of soil macrofauna is extremely low in the maize fields we have sampled, the potential for a holistic approach is quite limited at present.
In my view the conclusion is opposite; the low level of biodiversity requires the prompt adoption of an holistic approach (conservation tillage practices, no prophylactic insecticide applications, organic matter incorporation, ……), making cultivation process sustainable and complying with European legislation;
Point 10: the following question has not been addressed: might generalist predators populations decrease other beneficial populations and make possible the outbreaks of other soil pests (e.g. you can control WCR larve but you may cause the outbreak of other maize soil pests due to the killing of their antagonists by the introduced predators.
It seems that we cannot exclude this occurrence; what can you say ..in a holistic approach?
Fig. 2: number of WCR beetles per plant
A language revision of the text, namely the additional sentences, is suggested;
I would suggest that you take your time and read the text carefully again, before and after language revision; some hours now can avoid that mistakes will be there forever ……..
Author Response
find the response to reviewer in the file attached

Reviewer 2 Report
The comments and suggestions is in the PDF attached.
Author Response

(The authors gave the same response as above.)
